# Drought Tolerant Varieties of Common Beans (*Phaseolus vulgaris*) in Central Afghanistan

**Sayed Muhammad Baqer Hussaini [1], Roy C. Sidle [2,\*], Zaigham Kazimi [1], Aziz Ali Khan [2], Abdul Qayum Rezaei [1], Zahra Ghulami [1], Taher Buda [1], Rahmatullah Rastagar [1], Ali Aqa Fatimi [1] and Zahra Muhmmadi [1]**

[1] Bamyan Center, New Campus, Bamyan University, Bamyan 1601, Afghanistan; smb_hussaini@yahoo.com (S.M.B.H.); s.z.kazimi@gmail.com (Z.K.); qayumrezaei@yahoo.com (A.Q.R.); gholamizahra012@gmail.com (Z.G.); bwdamhmdtahr@gmail.com (T.B.); rahmatullahrastagar1@gmail.com (R.R.); bu.web@bu.edu.af (A.A.F.); z.muhmmadi@yahoo.com (Z.M.)

[2] Mountain Societies Research Institute, University of Central Asia, 155Q Imatsho Street, Khorog GBAO 736000, Tajikistan; azizali@ucentralasia.org

\* Correspondence: roy.sidle@ucentralasia.org

**Abstract:** Legume crops have played a significant role in the historical dietary regime of Afghan peoples. Recently, production of common beans has increased on Afghan farms relative to other leguminous crops. However, compared with other pulse crops, common beans are more prone to water stress. To select drought resistant common beans, several varieties were assessed in the field during a sequence of restricted water supplies for two years and the local drought regime was analyzed for a 12-yr period. The first experiment in 2018 compared five bean varieties under four irrigation regimes. White and black beans with long maturation periods and climber habits, and motley beans, characterized by moderate maturity and semi-climber structures, were susceptible to drought and did not mature well under restricted irrigation and ambient climate conditions. The other two varieties, red and pied beans, adapted to restricted water supplies and the long dry summers; these two varieties were assessed again in 2019. Statistical analyses and inferences based on the 2019 study suggest that red beans are more adaptable to water deficit treatments compared to pied beans. Therefore, red beans are considered a better option given the frequent mid- to late-summer droughts that occur in this region, together with the generally harsh mountain climate and short growing season of the central Afghanistan highlands. As a second varietal choice, pied beans are reasonably drought tolerant based on our findings.

**Keywords:** varietal selection; drought stress; common bean; climate patterns; restricted irrigation; Bamyan; Afghanistan

## 1. Introduction

Drought stress is one of the most important factors limiting crop growth and productivity, particularly for legumes in arid and semi-arid regions [1–3]. Considerable research has focused on efforts to improve drought tolerance and production of common beans through selection of various physiological and genetic traits; however, most of these investigations have been conducted in Latin America and parts of Africa [4]. While climate change is adversely affecting future drought projections in many areas, effects on common bean production and nutritional quality vary across the drier regions of the world [5,6]. Projected increases in drought occurrences and resultant impacts on yields of both rainfed and irrigated crops have major consequences for food security, particularly in developing nations, requiring appropriate adaptation measures [7–9]. Considering the declining groundwater levels in many dry regions due to consumptive water use, further unsustainable withdrawals for alleviating drought impacts on crops need to be carefully assessed [10].

Common beans (*Phaseolus vulgaris* L.) are the most important food legumes, providing the major source of dietary protein, vitamins, and minerals in many developing countries [11,12]. In many areas, increasing competition for production of other agricultural crops has shifted bean cultivation to marginal zones often associated with high abiotic stresses [13]. Drought stress is a worldwide production constraint of common bean production [14–16], both in seed and pod numbers [17–19], resulting in seed yield reductions of up to 60% globally [20]. Thus, drought tolerance is becoming a key trait for selection of common bean production in regions prone to drought stress due to diminishing water supplies, climate change, shifts in production areas, and increasing input costs [21]. While genetic improvements in common beans have benefited drought resistance, matching drought adaptation traits with local environmental attributes (e.g., drought patterns, growing season, temperature regime, soil fertility, pathogens) remains a challenging task [4]. Dry soils have high negative matric potentials ($\psi_m$), which restrict root uptake of water and transport through plant tissues; thus, osmotic potential ($\psi_s$) and turgor potential ($\psi_P$) of beans can be effective selection traits for drought tolerance screening [22]. Stomatal control of transpiration losses can be an effective adaptation trait for beans grown under drought stress conditions [23]. Lack of water has a negative effect on plant physiology (especially photosynthesis), and, if this water shortage continues, crop growth and yields are greatly reduced [24].

While overall agriculture and connected livelihoods in Afghanistan suffer from an arid to semi-arid climate with very little rainfall in the summer growing season, drought in the past few decades has worsened these conditions, particularly in high elevation regions [25]. These droughts have exacerbated historic dietary problems that exist among Afghan families, particularly in children [26]. Harsh drought in Bamyan caused a nearly 50% decrease in cultivated areas in 2014 compared with roughly a 15% decline in 2004 [27]. Due to these long-term water deficits, it is important to test for the drought response of multiple common bean varieties over time and under both stress conditions and reduced stress treatments [13,18,28,29]. Therefore, this study was designed to identify the varieties of the common bean that are tolerant to the dry mountain climate in Bamyan, as well as to restricted irrigation treatments. There have been no assessments of drought tolerant bean species nor drought patterns in this region**.**

## 2. Materials and Methods

This study was conducted in Yakawlang district of Bamyan province, Afghanistan, latitude 34°44′25″ N and longitude 66°57′01″ E, at an altitude of 2538 m during the 2018 and 2019 growing seasons (Figure 1). This area lies on the western end of the Hindu Kush Mountain range characterized by seasonally lush valley vegetation bordered by steep mountain slopes with little vegetation and numerous rock outcroppings. Soil at the experimental site was sampled and tested by the Soil Science Department of Bamyan University. Soil textures ranged from sandy loam to loam (Bouyoucos method), average pH was 8.02 (measured with a pH electrode in 1:1 soil:water slurry), and electrical conductivity (EC measured as the electrical resistance in a 1:5 soil:water suspension) was relatively low (0.78 mS m$^{-1}$). Climate in the area is transitional between cold arid and semi-arid, with cold winters and relatively warm, dry summers.

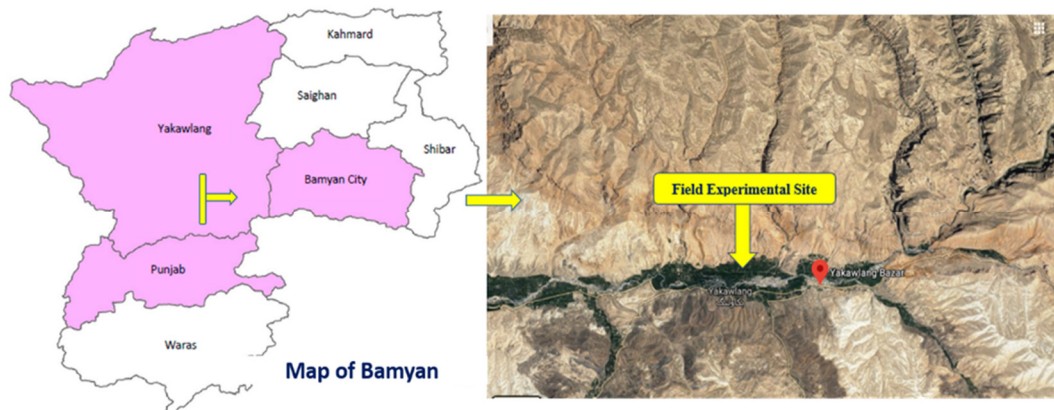

**Figure 1.** Location of experimental site in Yakawlang district of Bamyan province, Afghanistan.

Based on incomplete temperature records from 2012 to 2017 at the nearby Yakawlang climate station (≈ 2.5 km from our study area), average temperature during the growing season was 10.0 °C, but diurnal fluctuations during mid-summer months were commonly in the range of 10 °C to 15 °C. Mean annual temperature is reported to be 7.4 °C [30]. May is the coldest month of the growing season with an average temperature of 6.1 °C and minimum and maximum daily temperatures of −8.6 °C and 29 °C, respectively. July is the warmest month with an average temperature of 12.1 °C and minimum and maximum daily temperatures of 0.8 °C and 30.6 °C, respectively. Much of the precipitation occurs during late winter and spring; annual precipitation was 192 and 265 mm in 2018 and 2019, respectively, of which 45% occurred as snow. From 2009 to 2020, mean annual precipitation was 236 mm, ranging from 144 mm in 2012 to 310 mm in 2015, based on data from Yakawlang climate station. The maximum monthly precipitation occurs from February to May, with March and April typically having the highest totals. Precipitation in March is mostly snowfall, but snow can also fall in April.

Field experiments were conducted in two successive growing seasons at the same site: May to September 2018 and 2019. In both years, we used a randomized complete block design (RCBD) to account for any spatial effects of environmental attributes in the field. In 2018, four treatments (I1–I4) consisting of various irrigation intervals (7, 10, 13, and 15 days) were used during which approximately the same amount of water (≈ 0.1125 m$^3$) was applied to each plot via basin irrigation with furrows. Within each plot, water was delivered via four furrows spaced 30 cm apart. Five local bean varieties (black, white, motley, pied, and red; V1–V5; Figure 2) were tested and were randomly assigned in the experimental design. In each plot, 40 bean plants were established and there were four replications (blocks) of all treatment combinations. As such, the RCBD was 4 × 5 × 4, a total of 80 plots (Figure 3). In 2019, the same experimental design was employed, except that only the two best performing bean varieties were tested; thus, the RCBD was 4 × 2 × 4, a total of 32 plots.

Primary plowing was conducted with a handy three-wheel small chines tractor (Wuxi Changfu Tractor Manufacturing Co. Ltd., Wuxi City, China), followed by manually forming the plot and its borders based on the furrow irrigation setting. Individual plot size was 2.25 m$^2$ (1.5 m × 1.5 m) and plots (as well as blocks/replications) were separated from one another by 1.2 m to prevent water infiltration into adjoining areas. Likewise, water ingress was controlled by laying plastic sheets (polyethylene films, 200-micron thick) under water channels between plots and replications.

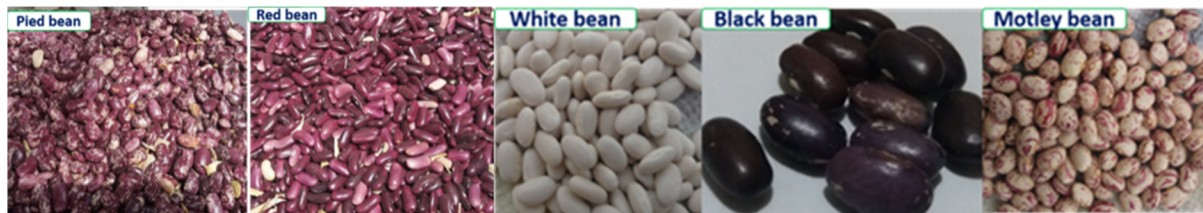

**Figure 2.** Seeds of five bean varieties planted in this study (from left to right): pied bean, red bean, white bean, black bean, and motley bean.

| Experimental design : RCBD in SPLIT PLOT | | | | | Main plot = Varieties = V | | | Sub Plots= Irrigation =I | | |
|---|---|---|---|---|---|---|---|---|---|---|
| Number of V= 5 | | | Number of I = 4 | | Number of R= 4 | | | Total plots=80 | | |
| | | | | | | North | | | | |
| 10 | 9 | 8 | 7 | 6 | 5 | 4 | 3 | 2 | 1 | |
| I4V4 | I1V5 | I3V5 | I1 V4 | I1 V3 | I2V4 | I3 V3 | I4 V1 | I3 V1 | I2 V2 | Rep 1 |
| 20 | 19 | 18 | 17 | 16 | 15 | 14 | 13 | 12 | 11 | |
| I2 V1 | I3 V4 | I2 V5 | I1 V1 | I4V3 | I1 V2 | I4 V5 | I3 V2 | I2 V3 | I4 V2 | |
| 30 | 29 | 28 | 27 | 26 | 25 | 24 | 23 | 22 | 21 | |
| I4V3 | I3V1 | I2V5 | I1V2 | I2V2 | I3V4 | I4V5 | I4V2 | I1V4 | I2V3 | Rep 2 |
| 40 | 39 | 38 | 37 | 36 | 35 | 34 | 33 | 32 | 31 | |
| I4V1 | I3V5 | I1V5 | I2V1 | I4V4 | I1V1 | I1V3 | I2V4 | I3V2 | I3V3 | |
| 50 | 49 | 48 | 47 | 46 | 45 | 44 | 43 | 42 | 41 | |
| I4V3 | I3V3 | I3V5 | I2V5 | I1V4 | I2V3 | I1V2 | I2V1 | I4V2 | I3V1 | Rep3 |
| 60 | 59 | 58 | 57 | 56 | 55 | 54 | 53 | 52 | 51 | |
| I4V5 | I1V3 | I2V4 | I4V1 | I1V1 | I3V4 | I3V2 | I4V4 | I1V5 | I2V2 | |
| 70 | 69 | 68 | 67 | 66 | 65 | 64 | 63 | 62 | 61 | |
| I4V4 | I3V3 | I1V5 | I2V3 | I2V1 | I1V3 | I2V4 | I3V4 | I3V1 | I4V2 | Rep 4 |
| 80 | 79 | 78 | 77 | 76 | 75 | 74 | 73 | 72 | 71 | |
| I3V2 | I4V5 | I1V1 | I2V5 | I1V4 | I4V1 | I1V2 | I4V3 | I2V2 | I3V5 | |

**Figure 3.** Field layout of the randomized complete block experimental design (RCBD), 4 × 5 × 4, employed in 2018 (I = irrigation interval, V = variety, Rep = replications, and numbers = plot numbers). Irrigation intervals are: I1 = 7 days (control); I2 = 10 days; I3 = 13 days; and I4 = 15 days. Bean Varieties are: V1 = black; V2 = white; V3 = motley; V4 = pied; and V5 = red.

In both years, crops were sampled at maturity, and plant height, grain yield, dry matter, number of pods per plant, number of stems per plant, number of seeds per pod, and hundred-grain weight were assessed (Figure 4). Thereafter, samples were dried in a laboratory oven at 75 °C for 72 h. Dry matter weight and other measurements were conducted, and data were statistically analyzed using Analysis of Variance (ANOVA) (Table 1); the LSD test was used to compare means.

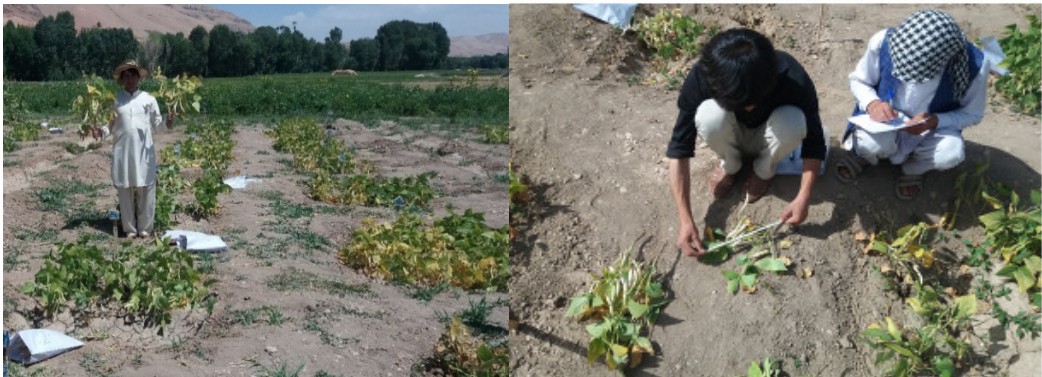

**Figure 4.** Field data collection in the experimental field site.

To cast our growth and yield findings into a broader climatic perspective, we quantified both the long- and short-term drought conditions based on 12 yr of daily precipitation data (2009–2020) from Yakawlang climate station. Given that only very incomplete temperature records were available and the precipitation time series is rather short to calculate most of the standard drought indices, we estimated long-term drought patterns based on normalized monthly average precipitation calculated as a 3-month moving average starting in March 2010. Short-term drought stress was estimated as daily rainfall based on 7-day moving averages from May through September of each year (i.e., the growing season).

**Table 1.** Analysis of variance for plant growth parameters of bean genotypes in irrigation treatments during the 2018 growing season.

| Sources of Variation | Df | SPP | HGWt | Yield | P Height | Pod No | Dry Wt. | Stem No | SPPd |
|---|---|---|---|---|---|---|---|---|---|
| | | | | Mean Square | | | | | |
| Replication | 3 | 29.1 | 92.8 | 17.9 | 455.2 | 4 | 66.6 | 2.5 | 2.6 |
| Variety (Vrty) | 4 | 1862.5 ** | 352.1 ** | 388.3 ** | 35605.8 ** | 645.6 ** | 1356.7 ** | 30.7 ** | 8.9 ** |
| Error (a) | 12 | 48.5 | 48.9 | 13.6 | 183.6 | 2.8 | 29.8 | 0.9 | 2.3 |
| Irrigation (Irr) | 3 | 233.9 ** | 8.2 ns | 92.9 ** | 916.8 ** | 65.2 ** | 254.9 ** | 0.7 ns | 1 ns |
| Vrty x Irr | 12 | 46.6 ns | 98.1 ns | 24.4 ** | 248.5 ns | 12.7 ns | 55.9 ns | 0.8 ns | 2.1 ns |
| Error (b) | 45 | 49.6 | 99.0 | 10.3 | 257.5 | 10.5 | 41.5 | 0.472 | 1.7 |

ns: Non-significant; **: significant at 1% probability levels, respectively. SPP = number of seeds per plant; HGWt = Hundred grain weight (gram); Yield = grain yield (gram/plant); P Height = plant height (cm); Pod No = Number of pods per plant; Dry Wt. = dry matter (gram/plant); Stem No = number of stems and branches; SPPn = number of seeds per pod.

## 3. Results

In 2018, maturation period increased for pied, red, motley, black, and white bean varieties, respectively. White and black beans, and to a lesser extent the motley beans, matured late because their climber growth characteristics require a long growing season. In contrast, red and pied varieties matured earlier since their bushy morphology supported a short growing period (Figure 5). The number of seeds per plant were significantly different among bean varieties and irrigation intervals. Significant differences among bean varieties were also found for other measured growth variables—hundred grain weight (HGWt), grain yield, plant height, number of pods per plant, dry matter production, number of stems per plant, and number of seeds per pod. In addition, there was a significant interaction between varieties and irrigation intervals on grain yield production of the plants (Table 1).

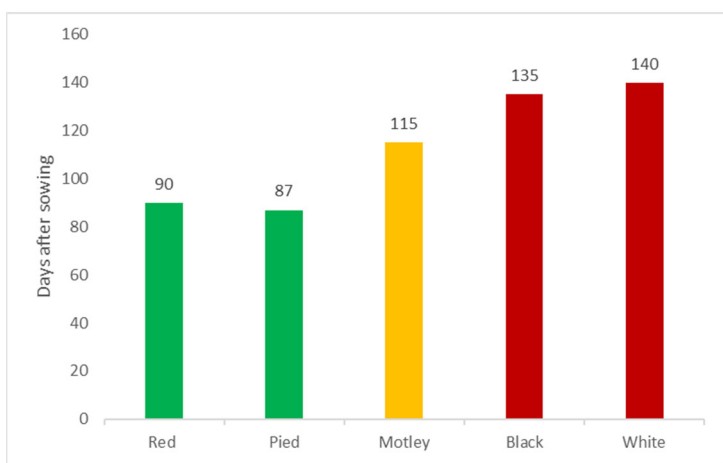

**Figure 5.** Maturity periods for the five bean varieties in 2018.

White, pied, and red beans had significantly more stems per plant than black and motley beans (Figure 6). White, black, motley, red, and pied beans were progressively shorter. Overall yields were highest for motley beans, followed closely by black, red, and pied beans, while white beans had significantly lower yields than all other varieties (Figure 6). Motley beans had greater hundred seed weight than both white and red beans. White beans plants had the highest dry weight of all varieties (Figure 7).

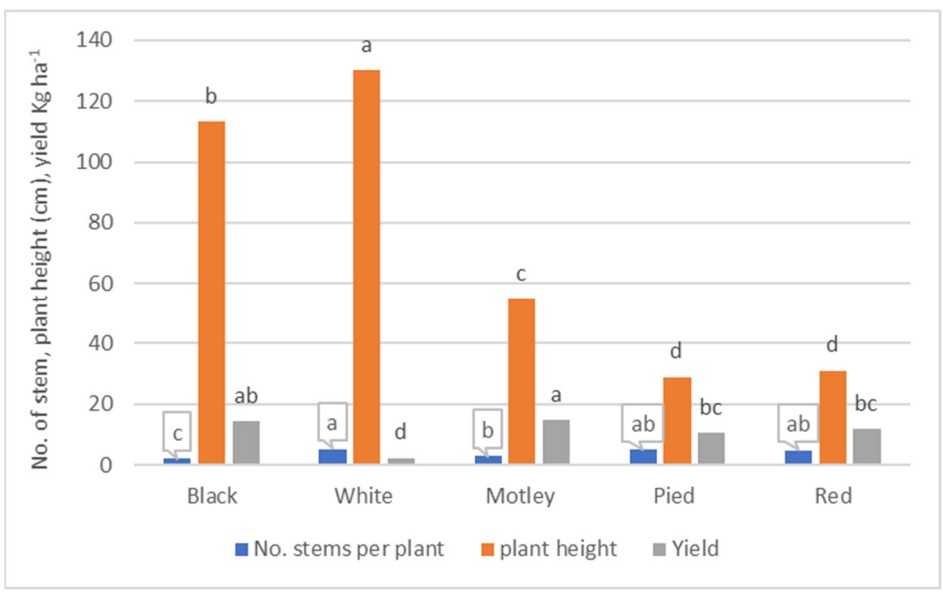

**Figure 6.** Mean comparison for number of stems per plant, plant height (cm), and grain yield (kg ha$^{-1}$) of plants under drought conditions (95% confidence interval). Overall standard errors for number of stems per plant, plant height, and yield are 0.34, 3 cm, and 1.5 kg ha$^{-1}$, respectively.

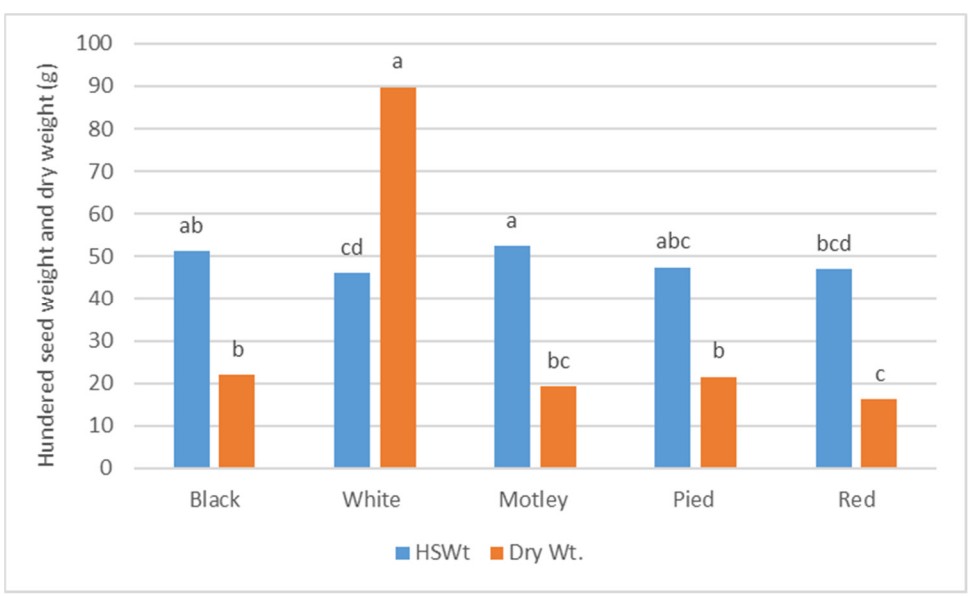

**Figure 7.** Mean comparisons among bean varieties for hundred seed weight (g) and dry weight (g) of plants grown under all combined water stress conditions (95% confidence level). Overall standard errors for hundred seed weight and dry weight are 1.3 g and 2.6 g, respectively.

White beans had significantly lower seed numbers compared to other varieties; black, motley, and red beans had the highest seed numbers (Figure 8). The number of matured pods per plant was highest for black beans and lowest for white beans. Immature pods were highest for white beans followed by black, motley, red, and pied beans (Figure 8).

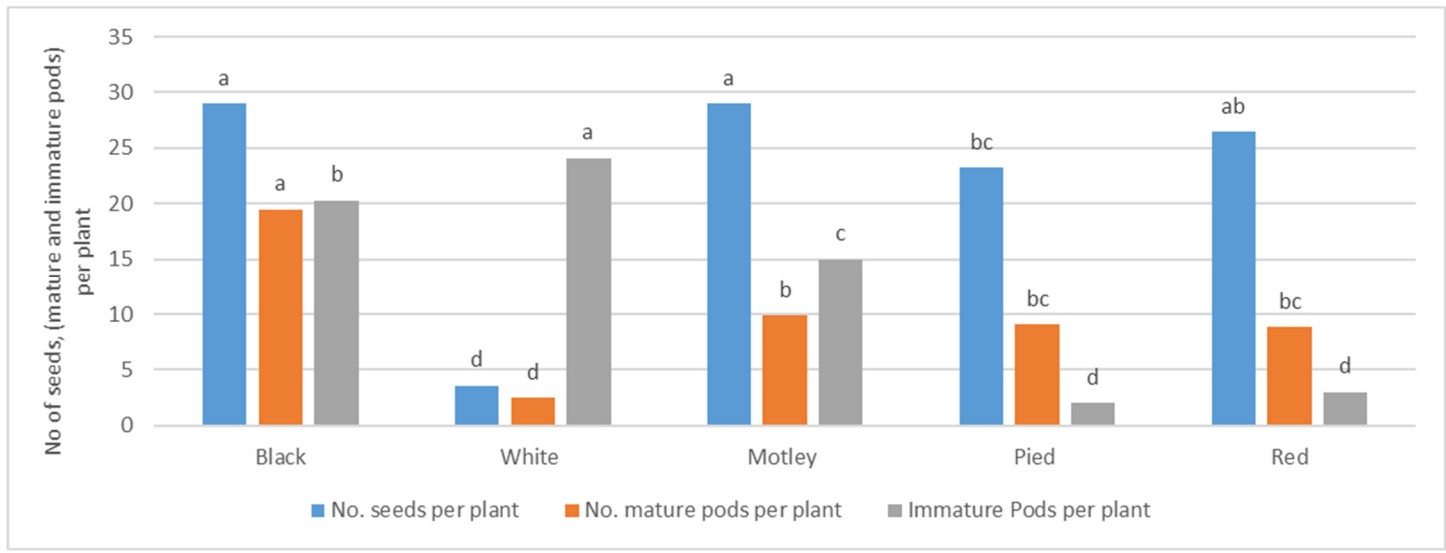

**Figure 8.** Mean comparison among varieties for number of seeds, and number of mature and immature pods per plant under all combined water stress conditions (95% confidence level). Overall standard errors for number of seeds, and number of mature and immature pods per plant are 2.2, 1.6, and 1.4, respectively.

Pied, red, motley, black, and white beans matured 87, 90, 115, 135, and 140 days after planting, respectively (Figure 5). The varieties that matured earlier (pied and red beans) physiologically adapted to dry conditions and were harvested earlier, thus not experiencing the late growing season water deficits. Motley beans matured later with some morphological susceptibility to dry conditions. Black and white beans matured considerably later, almost outside the growing season near the time of the first frost and during a period of no rainfall.

During the second growing season (May to September 2019) when only the best adapted beans (red and pied) were grown, both hundred grain weight and yield production were significantly higher for red beans (Table 2). Longer irrigation intervals significantly reduced grain yield production, plant height, number of pods per plant, and dry weight. Aside from the control irrigation interval (7 days), the 10-day interval was optimal compared to the 13- and 15-day intervals. The interaction of irrigation regime with varieties significantly affected the number of seeds per pod (Table 2).

**Table 2.** Analysis of variance on plant growth parameters of bean varieties in irrigation treatments.

| S.O.V | DF | Mean Square | | | | | | |
|---|---|---|---|---|---|---|---|---|
| | | HGWt. | Yield | P Height | Pod Number | Seed Per Pod | Dry Wt. | Stem Number |
| Replication | 3 | 18.3 | 70.3 | 25.3 | 8.7 | 0.6 | 245.5 | 8.326 |
| Variety | 1 | 186.8 ** | 140.2 * | 0.2 ns | 0.1 ns | 1.5 ns | 519.2 ns | 0.5 ns |
| Error (a) | 3 | 6.2 | 14.5 | 0.6 | 3.8 | 0.3 | 60.8 | 0.469 |
| Irrigation | 3 | 8.4 ns | 87.1 ** | 39.4 ** | 11.6 * | 0.1 | 309.5 ** | 2.8 ns |
| Variety x Irr | 3 | 16.9 ns | 34.3 ns | 11.ns | 2.4 ns | 0.5 ** | 153.8 ns | 1.7 ns |
| Error (b) | 18 | 31.3 | 12.2 | 9.9 | 3.2 | 0.1 | 55.5 | 1.109 |

ns: Not significant; * and **: Significant at 5% and 1% probability levels, respectively; SPPlnt = number of seeds per plant; HGWt = hundred grain weight; Yield = grain yield production; P height = plant height; Pod No. = number of pods; Dry Wt. = dry weight; Stem No. = number of stems and branches; and SPPod = number of seeds per pod.

Long-term drought stress was calculated based on monthly normalized precipitation (*NP*) as:x

$$NP = \frac{x - \bar{x}}{\bar{x}} \tag{1}$$

where $x$ is average precipitation for a particular month and $\bar{x}$ is the 12-yr average precipitation for the same month. Examining the drought patterns experienced during our two growing seasons in the context of the longer climate record, clearly there has been a long-term drought from late 2010 to 2020 except for 2015 to mid-2016 (Figure 9). Our study years (2018–2019) exhibited long-term drought stress below the 12-yr mean in almost all months except May 2018, January through March 2019, and May through June 2019. However, these brief periods were only slightly above the average normalized precipitation (see insert in Figure 9).

There was evidence of short-term drought stress during portions of each growing season in the 12-year climate record (calculated as a 7-day moving average of daily rainfall) (Figure 10). Drought patterns in both years of our field study were similar to those in 2014 and 2017. In these four years, the growing season started with somewhat moist conditions that would promote germination, but the entire mid- to late-summer period received almost no rainfall (Figure 10). The only years with greater short-term drought stress were 2011 and 2016, which both had little precipitation at the onset of the growing season followed by little to no precipitation in summer. The other six years had slightly better moisture distribution throughout the growing season, albeit some extended drought periods (Figure 10).

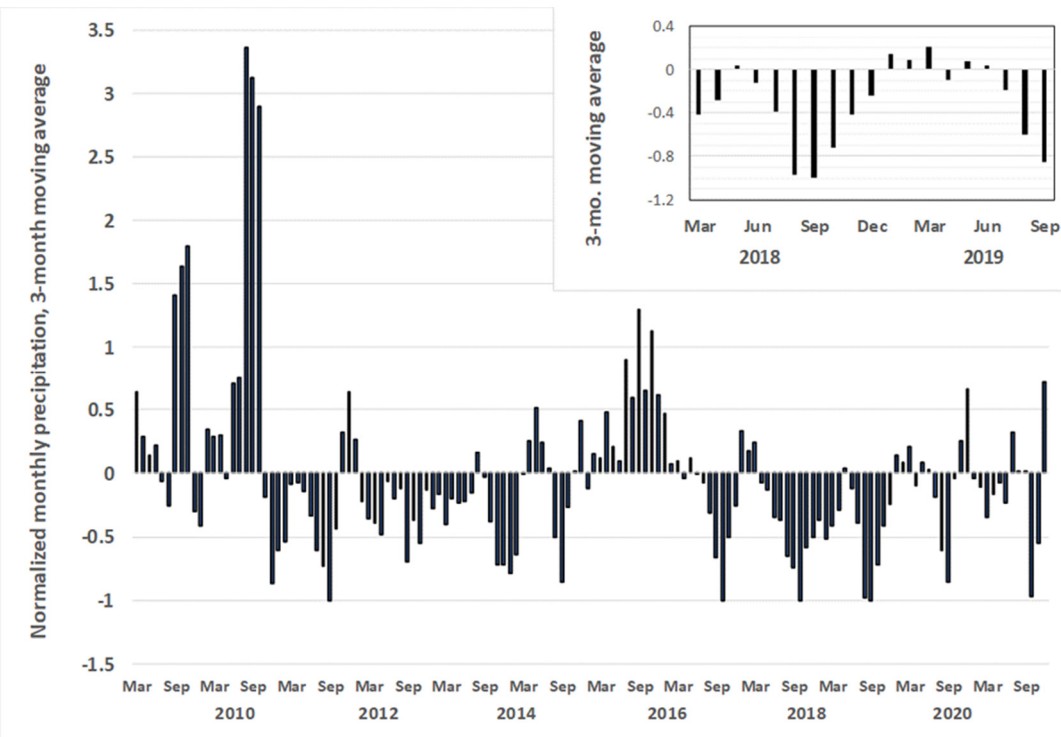

**Figure 9.** Normalized monthly precipitation calculated as a 3-month moving average from 2009 to 2020. Zero horizontal line represents the average monthly precipitation; deviations above and below this show months of higher precipitation and drought, respectively.

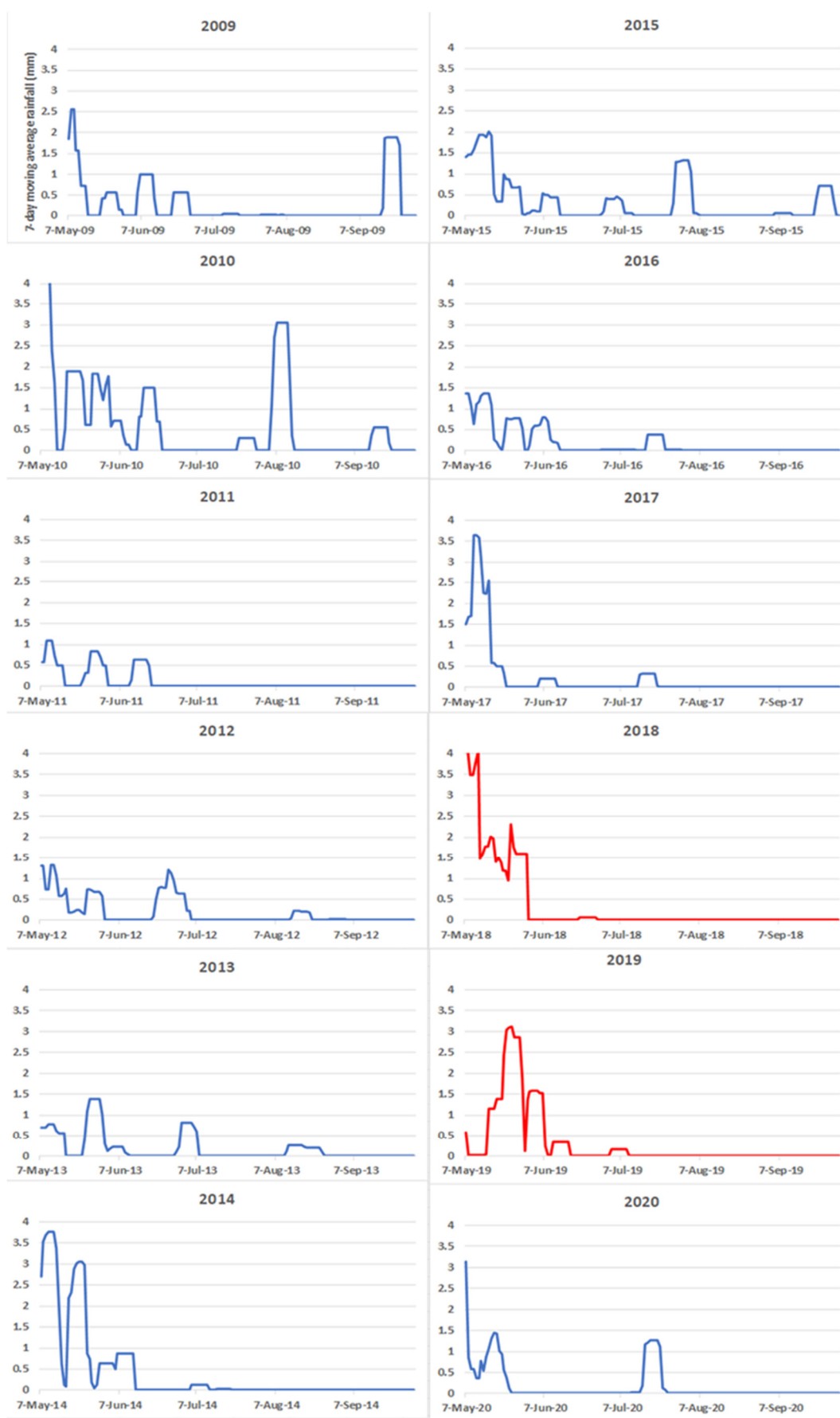

**Figure 10.** Short-term drought stress indicated by daily precipitation computed as a 7-day moving average for each growing season (May to September) from 2009 to 2020. Our two study years are noted in red plots (2018 and 2019).

## 4. Discussion

During the two-year field trials different varieties responded differently under stressed and non-stressed water conditions. These two years were part of a long-term drought cycle in central Afghanistan with very little rainfall during the mid to late growing seasons of both years (Figure 10). Half of the 12-yr climate record in this region exhibited severe drought stress during most of the growing season. Our findings show that grain yield reduction of plants occurred during periods with longer irrigation intervals affecting the number of pods, number of seeds per pod, and the dry matter weight of plants. High numbers of immature pods on white, black, and motley bean plants indicate that they had difficulty coping with water deficits and are quite susceptible to mid- to late summer drought. In contrast, the pied and red beans matured earlier and produced normal biomass while exhibiting suitable growth attributes. Taller plants (white, black, and motley beans) were more vulnerable to restricted irrigation regimes compared to shorter plants (pied and red beans).

White, black, and motley beans are more vulnerable to dry conditions because of their long maturation period. Late season drought vulnerability is very important in this area as nine out of the 12 years of climate record showed no rainfall in late August and early September, the time when late maturing bean varieties would be harvested. Plant productivity of white, black, and motley beans decreased 20–40% compared to the control (7-day irrigation interval) when water was applied in intervals of 10, 13, and 15 days, respectively, highlighting the drought vulnerability of these late maturing varieties. Alves et al. [31] reported a 33% decrease in bean productivity in Brazil where plants were subjected to 8-day irrigation intervals compared to 1-day intervals. In a water stressed environment in Iran, dry matter and grain yield of beans decreased about 40% during flowering and grain filling stages [32]. In Chile, green-shelled beans grown under optimal moisture conditions produced nearly 2- and 3-times the biomass compared to severe water deficit conditions (30% of optimal) during the vegetative and reproductive stages of development, respectively [18].

As black, white, and motley varieties require longer growth periods and are considered late maturing beans compared to pied and red beans, the former are not desirable varieties in this drought-stressed environment. Given that early maturity is a desirable agronomic characteristic in regions with short growing seasons that experience water stress, pied and red beans appear to survive and produce under drought conditions. Both red and pied beans reacted better to imposed water shortages compared to the other three varieties. Additionally, wilting occurred during water stress treatments in 5–10% of black and white bean plants and in 1–2% of motley bean plants. Wilting symptoms were not observed in these same irrigation treatments for pied and red beans. Comparing red and pied beans, red beans produced higher hundred grain weight (HGWt) and grain yields.

In accordance with our findings, Sadeghipour [33] showed that water shortages decreased dry weight and plant height of beans with the variety of D81083 more tolerant to water deficit conditions compared to Akhtar and Derakhshan varieties. For pinto beans grown in Iran, hundred grain weight, biomass production yield, grain yield, and number of pods per plant were negatively affected by water stress [34]. As our study site was in a mountainous (2500–2600 m a.s.l.) dry climatic zone, it is understandable that such conditions decreased the seed yield of bean varieties; however, even in tropical and sub-tropical environments, low red bean seed yields have been reported during periods of high evapotranspiration [35–37]. White and Singh [20] reported a 60% seed yield reduction of common beans after early afternoon plant wilting caused by water stress. However, in our study, wilting only occurred on climber varieties like white, black, and motley beans under restricted irrigation treatments.

Since only antecedent rainfall prior to planting is known for each growing season, given that at least half of the 12-year record in Bamyan had very dry summers, and 75% of the years had no rainfall late in the growing season, red and pied beans are better adapted to typical drought stress conditions compared to black, white, and motley beans.

During the second year of our study, red beans proved superior to pied beans. Thus, red and pied beans could be used as a baseline comparison against other varieties with respect to drought tolerance in similar mountainous environments.

## 5. Conclusions and Recommendations

Our findings revealed that white, black, and motley beans are long growth and late maturing plants, while red and pied beans mature much earlier under the same dry and high elevation conditions. Since early maturity is a key advantage of drought resistant plants, white and black beans may not be suitable for mountainous dry conditions, as both are fully climber plants. Motley bean was the most productive variety with moderate maturity period and a semi-climbing manner, and it can be a productive genotype in dry mountainous areas with adequate water supplies. However, white, black, and motley beans experienced grain yield reduction under restricted irrigation regimes and are thus highly susceptible to droughts common during the latter part of the growing season. Red beans are more productive and adaptable to water stress compared to pied beans, but both these varieties are better adapted to restricted water supplies and water stress compared to white, black, and motley beans. Therefore, red bean plants are the best option given the dry and harsh climate patterns associated with this mountainous region of central Afghanistan, particularly when irrigation water is in limited supply. Pied beans are a somewhat drought tolerant variety. Both varieties should be considered as suitable genotypes for drought tolerance in the region. Additional field studies in such dry, high elevation mountainous conditions are needed to cope with these agronomic challenges.

**Author Contributions:** The listed authors contributed to the research as follows: S.M.B.H. conceptualized the field study and designed the field experiments with assistance from Z.K., A.A.K., and A.Q.R.; R.C.S. developed the drought assessment and analyzed these data; S.M.B.H., Z.K., A.A.K., A.Q.R., T.B., R.R., A.A.F., Z.G., and Z.M. conducted the field work; S.M.B.H. and Z.K. analyzed the bean growth and yield data; S.M.B.H. and R.C.S. prepared the manuscript; and A.Q.R., T.B., R.R., and A.A.F. reviewed parts of the original manuscript draft. All authors have read and agreed to the published version of the manuscript.

**Funding:** This research was part of the University of Central Asia's "Pathways to Innovation" project funded by International Development Research Centre (IDRC) in Canada and Aga Khan Foundation Canada (AKFC).

**Institutional Review Board Statement:** Not applicable.

**Informed Consent Statement:** Not Applicable.

**Data Availability Statement:** Any inquiries regarding bean growth and yield data can be made to S.M.B.H (Email: smb_hussaini@yahoo.com). The climate data are restricted as a requirement of the Afghanistan Government.

**Acknowledgments:** The Afghan authors are thankful to University of Central Asia (UCA) and Mountains Societies Research Institute (MSRI) for the extensive support in this research project and opening close academic relationships with Bamyan University toward our joint mission on the development of mountains societies in Central Asia. Moreover, we sincerely thank our colleagues at Bamyan University for their timely and important contributions to this study. Note: Because of the recent Taliban takeover in the Bamyan region, the Afghan authors were unable to submit this paper and asked R.C.S. to submit it. We appeal to common wisdom to allow this group at Bamyan University to continue their research and plead for their safety. Currently they have limited access to their university files and materials.

**Conflicts of Interest:** The authors declare no conflict of interest. The funders had no role in the design of the study, in the collection, analyses, or interpretation of data, in the writing of the manuscript, or in the decision to publish the results.

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
