# Peer review of "Drought Tolerant Varieties of Common Beans (Phaseolus vulgaris) in Central Afghanistan"

_agronomy, doi:10.3390/agronomy11112181_

Round 1

Reviewer 1 Report

I would like to thanks the authors for their effort. The made some changes but still lot needs to improve. Details comments incorporated on edited manuscript. I want to mentioned two critical points here. 

Author didn't mentioned which irrigation treatment is suitable. Also author didn't discussed Table 1 in discussion chapter.

I would suggest to read couple of recent articles on drought resistant in legumes especially common bean and rewrite the manuscript.

Author Response

Please see attached response to Reviewer #1; the revised manuscript is being sent.

Reviewer 2 Report

According to my initial review of the manuscript

  • Although the Authors followed my suggestions and improved the structure of the Introduction section, I noticed that the last 2 paragraphs of Intro contain repeating information and should be reformed
  • Lines 31-34: You are saying that “Droughts and extreme heat are estimated to have reduced national cereal production by 9-10% globally…”. I am confused with the usage of the words national and globally. Please clarify
  • Ι recommend that you remove photos from Figure 3 and Figure 4, since they do not provide any valuable information
  • I repeat my statement for the Results section since the Authors only generated small improvements. The Results section of a scientific paper should simply state the findings, without bias or interpretation, and should be arranged in a logical sequence. Instead of the aforementioned, the Authors decided to present their results in the form of an endlessly repetitive data, failing to communicate this complex information as clearly and precisely as possible
  • In line with my initial comments, I am convinced that the statistical analysis in this paper is not correct. For example, focus on plant height (Figure 6). White is characterized as “a”, Black as “b”, Motley as “c”, Red (whose value is bigger than Pied) as “de” and Pied as “d”. This is an erroneous statistical presentation of the results, which raises questions about the validity of your conclusions. Suppose that this is a typing error, and instead of “d” you wrote “de”. However, similar errors are repeated in Figure 7 and 8. Consult an expert and statistically interpret your results from the beginning, since I believe that this is the main disadvantage of the manuscript

Author Response

Please see attached response to the reviewer's comments; the revised manuscript will be submitted.

Reviewer 3 Report

The manuscript contains very interesting, original and useful data.  It is generally well organized. The results are clearly presented and discussed. However, some points should be corrected.

The introduction does not provide sufficient background, e.g., the impact of drought on the cultivation of beans, and the physiological adaptation of beans to drought stress conditions are not adequately described.

Figure 3 is of low resolution. Please provide this FIgure in higher resolution.

The caption for Figure 3 should be more detailed. Please add irrigation levels and names of varieties.

Line 129: Have the assumptions for ANOVA been checked? What tests were used?

Author Response

(The authors gave the same response as above.)

Round 2

Reviewer 1 Report

Comments added with edited manuscript. Standard error needs to incorporated in figure or figure change to table.

Author Response

This comment arrived late after I contacted the Bamyan, Afghan senior author. Basically this issue was previously addressed when we added standard error data in the figure captions and now we have revised a few erronous mean seperations shown in Figures 6 and 8.

Reviewer 2 Report

Although, the authors managed to answer to the majority of my comments, i still detect erroneous statistical presentation of the results. For example, the five orange bars in Figure 8 are classified as: a-b-bc-bcd-e. Where is the "d" bar? The five grey bars in the same Figure are classified as: a-b-c-d-de. I hope that you realize that something is going wrong with the statistical classification of the last two values. Please, doublecheck the statistical interpretation of your data and be 100% certain about the presentation of your figures

Author Response

I apologize for mistakes made in Figs. 6 & 8 regarding the mean seperations. I did not have these raw data but I was finally able to contact the Afghan senior author in Bamyan who said - "The Figures 6 and 8 are now edited based on the last comments and remarks of the reviewer and your  emphasis as in excel file attachment. We are very sorry about this repeated mistakes and embarrassing edition by my side due to here conditions,  however still we are not in calm thinking and concentrations. Hope to accept my apology and find well the attachments."

I have checked the mean seperations in Figures 6 & 8 and they apprear to be correct. Thank you for catching these mistakes. The revised Figures are uploaded in the paper that is resubmitted. 

This manuscript is a resubmission of an earlier submission. The following is a list of the peer review reports and author responses from that submission.

Round 1

Reviewer 1 Report

First of all, I would suggest consulting an English language expert and checking the manuscript for grammar and syntax errors.

Additionally, I have the following minor and major concerns about the manuscript

  • The Introduction writing style is like reading a telegraph message. Short sentences that state or imply that the authors are unsure about themselves, are not attentive to detail, or could not find the right word and use a close approximation. Poorly written Introduction that lacks important information, and conveys a biased picture of the topic. A well-structured literature search is more than necessary
  • Line 30: Delete the word “national”
  • Lines 32-37 (“Increased crop yields…water tables”) contain isolated and unconnected pieces of information. I keep reading the text over and over again, trying to figure out what the authors are trying to say. Rephrase your statements and clarify
  • Delete or rephrase Lines 46-49 (“Water deficits…grain production”) and avoid repeating information given previously
  • Line 50, consider using the term “yield” instead of the term "production”
  • Materials and Methods: Describe the methodology used for the determination of soil texture, pH, and EC
  • It would be better for the reader to present data in Lines 82-95 (“Based on…in April”) as a Table
  • Correct me if I am wrong, but reading the Materials and Methods section, I see no description of your treatments. Which were the different irrigation levels applied? What do the abbreviations I1V1, I1V2, etc. in Table 3 stand for? What was the methodology used to evaluate that plants actually imposed drought stress? How many plants were growing in each plot? I am sorry, but your experimental procedure is unclear.
  • The Results section of a scientific paper should simply state the findings, without bias or interpretation, and should be arranged in a logical sequence. Instead of the aforementioned, the Authors decided to present their results in the form of endlessly repetitive data, failing to communicate this complex information as clearly and precisely as possible
  • Lines 118-119: Poor description of the statistical methods used to analyze data. Moreover, I am convinced that the statistical analysis in this paper is insufficient. For example, check Figure 7. The last green bar is characterized as “be”, the fourth blue bar is characterized as “abd” etc. Consult an expert and statistically interpret your results from the beginning
  • The discussion of the results is poor, containing elementary knowledge on the topic, together with the incredibility of the statistical findings and of the experimental design used, are by far the main disadvantages of this manuscript

Reviewer 2 Report

I would like to thanks the authors for their hard work. But the experiment is poorly designed. Only five genotypes and two years of study (2nd year only two genotypes) can't explain the drought impact. The phenology of all five genotypes is different, so it's obvious that they have different genotypic effects. Drought is a complex trait, and identifying genotypic/environmental/ genotype X environment affects the experiment needs a proper design. Irrigation treatment is not mentioned elsewhere in the manuscript. The graph should be improved according to the suggestions in the edited manuscript. The discussion part was not clearly explained and not supported with relevant references.  

I would suggest to the authors rewrite the manuscript based on phenology, growth pattern, and yield of common bean under mountainous and dry environments.